# Furloughed Employees’ Voluntary Turnover: The Role of Procedural Justice, Job Insecurity, and Job Embeddedness

**DOI:** 10.3390/ijerph20095664

**Published:** 2023-04-27

**Authors:** Felix Ballesteros-Leiva, Sylvie St-Onge, Marie-Ève Dufour

**Affiliations:** 1Department of Management, Université Laval, Québec, QC G1V 0A6, Canada; marie-eve.dufour@fsa.ulaval.ca; 2Department of Management, HEC Montréal, Montreal, QC H3T 2A7, Canada; sylvie.st-onge@hec.ca

**Keywords:** furloughed employees, voluntary turnover, procedural justice, job insecurity, job embeddedness

## Abstract

During the COVID-19 lockdown period, several employers used furloughs, that is, temporary layoffs or unpaid leave, to sustain their businesses and retain their employees. While furloughs allow employers to reduce payroll costs, they are challenging for employees and increase voluntary turnover. This study uses a two-wave model (Time 1: *n* = 639/Time 2: *n* = 379) and confirms that furloughed employees’ perceived justice in furlough management and job insecurity (measured at Time 1) explain their decision to quit their employer (measured at Time 2). In addition, our results confirm that furloughed employees’ job embeddedness (measured at Time 1) has a positive mediator effect on the relationship between their perceived procedural justice in furlough management (measured at Time 1) and their turnover decision (Time 2). We discuss the contribution of this study to the fields of knowledge and practice related to turnover and furlough management to reduce their financial, human, and social costs.

## 1. Introduction

The furlough of employees is a temporary layoff or unpaid leave that employers may implement during periods of economic uncertainty [1]. This human resource strategy allows employers to reduce labour costs while retaining talent so that operations can restart more quickly and easily when conditions improve [2]. Compared to other workforce reduction and job retention practices, furlough is the best practice to maintain the inversions, experience, and knowledge of a company’s human capital [2,3,4]. By avoiding permanent layoffs, employers can build goodwill and protect their company’s brand and reputation by demonstrating their commitment to their employees in difficult times [5], which might favor their loyalty, engagement, and performance [3]. In addition, furloughs allow employers to avoid losing their investments in recruitment, training, development, and performance management activities.

During furloughs, employees generally receive benefits and may be eligible for government programs such as employment insurance. However, they are not allowed to work or receive a salary from their employer during this period. In Canada, the Federal Government implemented the Canada Emergency Wage Subsidy (CEWS) program to support businesses impacted by the COVID-19 pandemic until May 2022. Eligible employers could receive, for 24 weeks, an assistance of up to 75% of wages paid, up to a maximum of CAD 1129 per week per employee. Several large Canadian companies have temporarily reduced their workforce to deal with uncertainty due to lockdowns and the pandemic. For example, in the spring of 2020, Air Canada and Canadian Tire furloughed around 16,500 and 17,000 employees due to the sharp decline in traveling demand and the closure of retail stores during the pandemic. Many governments have implemented furloughs during budget crises [1,6,7].

While furloughs allow employers to reduce payroll costs while retaining their talents, they can be challenging for the employees involved. Furloughed employees generally receive benefits, and those who may be eligible for government programs such as employment insurance are not allowed to work or receive a salary during their leave. During the furlough periods, they may experience financial difficulties, loss of support, status, and pride, and suffer stress, anxiety, and depression [7,8,9]). Similarly, prolonged leave can lead to a decline in employee morale and satisfaction, especially for those who feel they are being treated unfairly compared to colleagues who were not furloughed (Lee and Sanders [5]. Furloughs also lead to an erosion of trust or justice perceptions given the breach in psychological contract between an employer and its employees [2,7,10].

In today’s talent shortages that create competition to hire skilled employees, there is a need to further understand employee turnover [11]. In this study, we are interested in the impact of furlough on voluntary turnover, that is, on employees’ voluntary cessation of membership of their organizations [12]. Many furloughed employees stayed with their employers while others voluntarily quit or expressed the intention to leave [13,14,15], particularly among higher-performing employees. This study innovates in exploring furloughed workers’ turnover during the COVID-19 lockdown period, on which there is scant research [13,14,15]. Its purpose is better to understand the impact of furloughs on voluntary turnover decisions and investigate the psychological process behind them. Understanding turnover, particularly among furloughed employees, and how to prevent and control it is highly important to scientists and practitioners [16,17]. Voluntary turnover remains a significant organizational challenge, considering the loss of an employee’s professional skills and the severe and various financial and human costs associated with it. These costs include the time and money to recruit and select suitable candidates, the loss of knowledge and productivity, the reduction in customer satisfaction, the stress and overload for those who must compensate for the departure of colleagues, etc. [13,16,18,19]. Employees’ voluntary turnover can also has a contagion impact on other employees leaving [18,20], reducing team and organizational performance [14,15]. All these negative impacts explain why the last three decades have seen a growing interest in understanding how employees leave their jobs and why they do so [15,21,22].

This study addresses the need for research that “explores the role of turnover and retention in national policies designed to promote post-pandemic economic recovery” [15]. We innovate in using conservation of resources (COR) theory to investigate the relationships between furloughed workers’ perceptions of justice, job insecurity, job embeddedness, and their voluntary decision to leave their employers. Although furloughs have been an increasingly common strategy in the public and private sectors for a long time [6,7], there is relatively little research concerning how furloughed employees’ perception influences their actual decision to quit their employer, which is different than their turnover intention.

From a methodological point of view, we also use an innovative approach by investigating furloughed workers’ decisions to quit their employer during the pandemic lockdowns in Canada using a two-wave study design. If employees had left their employers at T2, they were then asked whether their departures were voluntary or involuntary. Those who had left for involuntary reasons (e.g., layoffs and firings) were excluded from analyses to focus on employees’ volitional quitting decisions. This focus on furloughed employees’ actual voluntary turnover appears to be value-added from previous research that mostly considers turnover intention, while the latter does not necessarily lead to actual turnover behaviour [15].

From a practical point of view, our results should help executives, managers, and human resources professionals better understand why furloughed employees quit their employers, allowing them to prevent and reduce their departures. Furloughs appear to increase the chance of losing employees, most often, the high-performing ones who possess the competencies to gain employment elsewhere, leaving a firm with mid to low performers [4]. Thus, through furloughing, employers risk losing more productive workers than if they decided to lay off the least productive workers [23]. Our results should provide insight into proactively avoiding the best employees’ leave during furloughs.

### 1.1. Theoretical Perspectives and Research Model

#### 1.1.1. Conservation of Resources (COR) Theory

The COR theory [24,25] explains human motivation and comportments based on a drive to preserve or conserve valued factors, otherwise known as individual, social, tangible, and symbolic resources. Its central tenet is that people strive to acquire, maintain, protect, and build resources to achieve their goals [24].

The concept of resources refers to “anything perceived by the individual to help attain his or her goals” [26]. It covers things people value, such as objects, conditions, personal characteristics, energies, and social support. Resources can be tangible or intangible, self-generated or derived externally, and held in different degrees across life domains [26]. In other words, resources are entities that have an instrumental value for individuals, such as objects (e.g., a house or car), conditions (e.g., being employed), personal resources (e.g., skills), and energies (e.g., mental and physical energy).

This theory suggests that employees with more resources have increased coping abilities and can better adjust to resource threats than those with fewer ones. Possessing significant resources increases people’s ability to solve problems and reduces the likelihood of having their well-being negatively affected by the draining of resources during stressful situations. Some principles and corollaries emerge from this strive for resources tenet [25,26,27]. Above all, the loss aversion principle states that, while both resource loss and gain are important to individuals when equal amounts of resources are involved, resource loss has a stronger impact on them than resource gain. Moreover, the resource caravan principle states that resources exist in aggregates: one may expect to obtain the value of a resource to the extent that it fits into one’s existing resource portfolio [26]. People invest resources to recover from a resource loss situation or to obtain additional resources. On the one hand, if they fail to gain and lose resources, they may enter a loss cycle in which the initial loss of resources leads to further losses. On the other hand, if people invest and gain resources, they may enter a gain cycle in which the initial gain leads to future increases.

The COR literature has identified several types of resource investment strategies to prevent further loss of resources or to enable their recovery [26]. Employees can use a cost–benefit analysis or a reasonable loss strategy before investing resources. There is also the exit strategy, such as a change to a less stressful job or leaving the organization, which is more likely to be adopted by employees with fewer resources, greater pessimism, or greater risk aversion [26,27]. A replacement strategy involves directly using existing resources to replace previously lost ones. Hobfoll (1989) uses the example of a laid-off person trying to find a new job using existing resources: skills, experience, social networks, industry knowledge, etc. [27]. Finally, employees may also re-evaluate a resource threat to cope with it. For example, they may conclude that a quick promotion may not be important if the work effort required in return results in higher domain-life conflicts. Based upon COR literature, we propose that employees placed on furlough status face the severe threat of resource loss both leading up to and during the actual furlough [4,9]. By linking this theory to voluntary turnover, it can be inferred that the loss and protection of resources during a furlough correspond to losses of tangible resources such as salary or benefits. Furlough also leads to losses of conditional resources, for example, organizational membership, social relationships at work, and personal resources such as self-efficacy and dignity. All these lost resources will affect employees’ behaviours and well-being and cause them to seek new work opportunities [2,28].

#### 1.1.2. Research Model

This study aims to explore the relationships in our research model illustrated in Figure 1. This section presents how COR theory and previous research led us to propose all of the hypotheses.

Based upon COR theory, justice is one of the primary sources of organizational support from which resources are accumulated, replenished, and protected [29]. Research shows that fairness concerns could affect employees’ attitudes and behaviours for reviews (see (for reviews, see [30,31])). When employees perceive being treated fairly by their superiors and employer, they feel compelled to reciprocate with actions that contribute to the organization’s goals [32,33]. Similarly, perceptions of unfairness will place harsh demands on employees, depleting valued resources and dissuading them from contributing to organizational goals [34]. Research indicates all types of justice perceptions have been linked meta-analytically to various outcomes, including satisfaction, commitment, citizenship, and withdrawal [35]. For example, employees who perceive that employers make decisions through fair procedures report higher levels of job satisfaction, organizational commitment, organizational trust, job performance, and lower turnover intention [30,36].

During furloughs, employees’ trust and sense of justice are negatively affected [7]. Procedural justice means employees expect employers to use fair processes and procedures when allocating work outcomes [37]. It then becomes particularly important to optimize perceptions of procedural justice in implementing and managing furlough, which is often enacted unilaterally rather than through bilateral consultation. Perceptions of furlough procedural fairness as resources can influence the employee’s adaptation to this temporary event and belief in a good return to work. Hamilton et al.’s qualitative study (2022) confirms the crucial role of communication interactions, including vocal communication, during a long furlough. Marginalization can stem from communication gaps that many furloughed employees experience. This negatively affects their perceptions of how furloughed employees have been selected and how they are treated during furlough. We then propose the following hypothesis based on COR theory:

**Hypothesis** **1:**
*Employees’ procedural justice perceptions of furlough management (at Time 1) are negatively related to their subsequent decision to voluntarily turnover (at Time 2).*


The unprecedented rise in furloughs during the COVID-19 pandemic increased perceptions of job insecurity, defined as the perceived threat of job loss and related worries [38], or, still, the perceived uncertainty surrounding future job prospects and feelings of powerlessness to control job retention [39,40,41]. Job insecurity can be perceived as a predictor of future job loss and implies unpredictability and uncontrollability [39]. The most detrimental aspect of job insecurity is the anticipation of and powerlessness to address the potentially negative work situation. It has also been shown that job insecurity has various harmful consequences on the satisfaction of individuals’ needs, personal identity, status, livelihood, well-being, mental health, organizational trust, and other work-related attitudes as well as their behaviours, such as their turnover [41,42,43,44,45].

COR theory [24,25] helps explain why employees who perceive more job insecurity are more likely to quit their jobs. This theory considers stable employment and job features, such as income and developmental opportunities, as resources [25]. It proposes two types of resource loss: potential loss and actual loss. It also explains why a perceived threat of resource loss might have a greater negative effect than an actual loss. When threatened with resource loss, individuals are likely to experience chronic stress, uncertainty, and helplessness, leading to greater psychological distress [24,25].

Conversely, job insecurity depletes employees’ resources through repetitive thoughts and worries about the future of their job. While furlough might help to maintain employment, it corresponds to involuntary job displacement that increases the feeling of insecurity and distress [7]. It even affects employees’ dignity or sense of self-worth or being recognized and appreciated by others [1]. Furloughed workers must face the uncertainty of returning to work where they would keep the same working conditions, professional development plans, or opportunities within their organizations. Their perceptions of job insecurity may lead to seeking job alternatives to withdraw from the stressful, uncertain situation. The higher the perceived number of other options, the easier it is for the furloughed employees to quit. When their careers are threatened, insecurity may further deplete employees’ resources. Employees may try to protect their remaining resources by withdrawing from a stressful situation in expressing turnover intention or quitting [44,45]. Richter et al. [45] confirmed that job insecurity, as well as ruminating about it, have a positive impact on actual voluntary turnover. Therefore, based on COR theory and the previous review, we propose that the more furloughed employees perceive job insecurity, the more likely they are to quit.

**Hypothesis** **2:**
*Furloughed employees’ perceptions of job insecurity (at Time 1) are positively related to their subsequent voluntary turnover decisions (at Time 2).*


Building on COR theory, Kiazad and colleagues [46] explain that employees maintain their status in a role (e.g., remain employed or remain invested in personal responsibilities) because they strive to accumulate and retain domain-specific resources. Once these resources are acquired/invested, they become difficult to relinquish—this is known as the primacy of resource loss [26,27]. Individuals have several levels of integration in different life domains (for example, as employees, family members, or community members) throughout their lives. This integration involves mechanisms of accumulation and investment of resources that vary over time in each life domain, coinciding with different levels of motivation to protect future resources to be invested in a particular life domain [35,47].

Job embeddedness includes organization and community embeddedness, each categorized into three independent components: links, fit, and sacrifice [48]. The links component refers to connections with work team members and colleagues that impose normative pressure on employees to stay in their job. The fit component involves an employee’s perception of their compatibility or comfort with the work environment, increasing their attachment to their job and organization. The sacrifices component captures the actual costs or potential losses from employees leaving their jobs. The more resources employees give up when they quit their jobs, the less attractive an external job opportunity will seem. Mitchell and colleagues [48] suggest that when employees feel more of these restraining forces (components), they become increasingly embedded in their current job and, consequently, are less likely to leave the organization voluntarily.

Kiazad and colleagues [46] also argue that the more resources a person has accumulated in each domain, the more integrated they are and the more reluctant they will be to give up domain-specific resources. The resources in the workplace can include recognition, skills, social support, or convenient schedules or status. Outside of work, resources can be expressed by enjoyable community activities, a safe and secure living environment, easy and convenient travel to work, or support from one’s spouse/neighbours [49,50]. People can apply these embedding resources to satisfy specific role demands. Still, their intrinsic or instrumental value can also buffer against undesirable role experiences, motivating people to remain in their roles to avoid resource losses [26,47]. To summarize, the previous review suggests a mediator effect of furloughed employees’ job embeddedness on the relationships between their perceived job insecurity and their perceived justice of furlough management, on the one hand, and their decision to quit their jobs, on the other hand.

**Hypothesis** **3:**
*Furloughed employees’ perceived justice of the furlough management (at Time 1) has an indirect effect on their subsequent voluntary turnover decision (at Time 2) via their job embeddedness.*


**Hypothesis** **4:**
*Furloughed employees’ perceived job insecurity (at Time 1) has an indirect effect on their subsequent voluntary turnover decision (at Time 2) via their job embeddedness.*


## 2. Research Methodology

### 2.1. Participants and Procedure

We collected data through an online survey of Canadian workers in 2020. Aligned with prior research on job embeddedness and turnover, (e.g., [51,52]), we used a research firm to access a large sample of respondents. We surveyed adults (age ≥ 18 years) through LegerOpinion, the largest Canadian web panel with 400,000 members across Canada. LegerOpinion ensures data quality through digital fingerprinting, background checks, and timestamps to flag careless responses. Similar online platforms provided evidence for the reliability and quality of the data collected [53]. We informed respondents of the purpose of the study, the voluntary nature of the survey, and the confidentiality of data treatment. Similarly, we also specified that the project was longitudinal, with additional questionnaires to complete over one year following the lockdown period in Canada. To constitute our sample of furloughed employees, we asked the research company to collect data from respondents that met the following screening criteria to ensure respondents had significant work and home responsibilities and to increase systematic variance in study variables: married women working full-time, having at least one child, spouse working full-time, having at least a four-year bachelor’s degree, and living in Canada. Participation in the survey was considered agreement to the terms (“informed consent”).

At T1, we contacted 7380 people across Canada. After a data-cleaning procedure, 639 responses remained from those who met the required furlough profile. We invited respondents who responded at T1 (*n* = 639) to participate in a follow-up study 6 months later (September 2020; T2 study or study 2). In total, 379 individuals responded at T2 (for an overall response rate of 59%). This result comes from the matching procedure between measurements at the two-time points that LegerOpinion conducted; we did not retain participants who could not satisfy this criterion in the final sample [54]. The sample included 297 male (46.5%) and 342 female employees (53.5%). The majority of them had a permanent work contract (77%). Eleven percent of the employees were under 25 years old, 46% were between 25 and 34 years old, 24% were between 35 and 44, 15% were between 45 and 54, and 6% were over 55 years old. Thirty-nine percent of the employees had a university degree. Respondents worked in thirty-five sectors: 11% in the retail commerce sector; 6.3% in the arts, entertainment, and recreation sector; 6.2% in the construction sector; 6% in the tourism sector; 5.9% in the health and social services sector; and 4.6% in the education and training sector.

### 2.2. Measures

#### 2.2.1. Voluntary Turnover Decision

At T2, we asked participants whether they still worked for the same employer. To do so, we measured voluntary turnover decisions with one dichotomous item, where participants indicated whether they had changed jobs to another organization during the furlough over the last three months (1 = yes, 0 = no). If they had left their organizations, we asked them to clarify whether their turnover was voluntary or involuntary with another dichotomous item. Participants who had left for involuntary reasons (e.g., layoffs and firings) were excluded from analyses to retain only those who made voluntary quitting decisions.

#### 2.2.2. Perceived Justice

At Time 1 or Study 1, we used the six-item global justice scale from Ambrose and Schminke [55] to measure perceptions of fair treatment in furlough management. Examples of items are: “[Regarding furloughs,] in general, the treatment I receive around here is fair” and “[Regarding furloughs,] overall, I’m treated fairly by my organization”. Participants had to answer on a scale from 1 (strongly disagree) to 5 (strongly agree). The scale’s coefficient alpha (α) is 0.93.

#### 2.2.3. Perceived Job Insecurity

We use the four-item scale from De Witte et al. [56]. One item is: “I feel insecure about the future of my job.” Participants must answer on a scale from 1 (strongly disagree) to 5 (strongly agree). The scale’s coefficient alpha (α) is 0.86.

#### 2.2.4. Job Embeddedness

We use the nine items scale developed by Crossley et al.’s [57] to measure job embeddedness. One of the items is: “I feel attached to this organization,” and the item “It would be difficult for me to leave this organization”. Participants must answer on a scale from 1 (strongly disagree) to 5 (strongly agree). The scale’s coefficient alpha (α) is 0.92.

#### 2.2.5. Control Variables

We controlled for three demographic variables (gender, age, tenure) and one perceptual variable (employability) based on prior research, e.g., (e.g., [45,58,59]). We coded gender: 0 for male and 1 for female. We measured age and organizational tenure by the number of years and perceived employability using the four-item scale developed by Rothwell et al. [60]. One of the items is “There is currently a strong demand in the job market for people like me”. Participants had to answer on a scale from 1 (strongly disagree) to 5 (strongly agree). The scale’s coefficient alpha (α) is 0.84.

### 2.3. Data analyses

#### Confirmatory Factor Analyses

We conducted structural equation modeling with Mplus 8.2. For the preliminary analyses (i.e., confirmatory factor analysis), we applied a maximum-likelihood (ML) estimation. We first conducted confirmatory factor analyses to examine the discriminant validity of our measures. We evaluated model fit using the comparative fit index (CFI; [61]), the Tucker–Lewis Index (TLI), and the root mean square error of approximation (RMSEA; [62]). For the CFI and TLI values, the traditional cut-off criterion is >0.90 [63], whereas the stricter criterion is >0.95 [64]. RMSEA values below 0.08 indicate an acceptable fit, whereas values below 0.06 are good [64]. A three-factor measurement model consisting of job insecurity, perception of fairness and justice, and job embeddedness demonstrated a satisfying fit to the data [64]: χ^2^ = 535, (114), *p* < 0.01, CFI = 0.95, TLI = 0.94, RMSEA = 0.07. The following alternative models had worse fits to the data: a one-factor model (χ^2^ = 4068.9, (119), *p* < 0.01, CFI = 0.54, TLI = 0.48, RMSEA = 0.23), and a two-factor model, in which job insecurity and justice loaded on one factor, and job embeddedness into another (χ^2^ = 1767, (118), *p* < 0.01, CFI = 0.80, TLI = 0.76, RMSEA = 0.15).

Because the dependent variable, voluntary turnover, was binary, we conducted a logistic regression model with an odds ratio indicating how likely an outcome of the dependent variable will occur, instead of the reference outcome, due to the change from an independent variable. We used the maximum likelihood estimator for measuring mediation based on Feingold et al.’s (2019) [65] approach. We assessed the statistical significance of indirect effects as the product of path A and path B using bias-corrected bootstrapped 95% confidence intervals (CIs; 10,000 draws). The bootstrap method produces standard errors, and confidence intervals can account for the non-normality of the distribution in small samples or the mediator, or the outcome is binary [65,66]. For the impact on the left side of the table (in the probability metric), relationships are statistically significant at *p* < 0.05 when the reported 95% CI does not include a value of 0. The respective ORs on the right side are statistically significant when the CI does not have a value of 1.

## 3. Results

### 3.1. Descriptive Statistics and Correlations

Table 1 shows the inter-correlations for all variables. Table 1 shows the inter-correlations for all variables. As expected, furloughed employees’ voluntary turnover decision measured at Time 2 is negatively linked to their perceived justice in furlough management (r: −0.181, *p* < 0.01) and positively related to their perceived job insecurity (r: 0.149, *p* < 0.05), both measured at Time 1. However, furloughed employees’ voluntary turnover decision measured at Time 2 is not significantly related to their perceived job embeddedness measured at Time 1 (r: −0.091, ns). Concerning control variables, furloughed employees’ employability measured at Time 1 is negatively related to their voluntary turnover decision measured at Time 2 (r: −0.254, *p* < 0.01). However, their voluntary turnover is not significantly associated with their age, gender, or education.

### 3.2. Hypothesis Testing

To test hypotheses 1 and 2, we used logistic regressions. Table 2 shows the logistic regression coefficients, their corresponding ORs, and 95% confidence intervals for these odd ratios. It also indicates the logistic coefficient effects in the probability metric (e.g., 0.2 means that the probability of y = 1 increases by 0.2 when x = 1, relative to x = 0). The results confirm hypothesis 1: perceived justice in furlough management at Time 1 is negatively linked to furloughed employees’ voluntary decision to quit at Time 2 (β = −0.40, *p* < 0.001). The results also confirm hypothesis 2: furloughed employees’ perceived job insecurity at Time 1 is positively linked to their voluntary decision to leave their employer at Time 2 (β = 0.23, *p* < 0.05).

Finally, the results presented in Table 3 confirm Hypothesis 3: Furloughed employees’ job embeddedness indirectly affects the relationship between their perceived justice in furlough management and their subsequent turnover decision (aOR = 0.953, 95% CI = [0.922, 0.979]). However, the results do not validate Hypothesis 4: Furloughed employees’ job embeddedness does not significantly mediate the link between their perceived job insecurity and subsequent decision to quit (aOR = 1.02, 95% CI = [0.987, 1.017]).

## 4. Discussion

### 4.1. Theoretical Implications

This study contributes to both the turnover and furlough literature domains. Concerning the knowledge area on turnover, our results align with theoretical models suggesting that organizational characteristics, such as management policies, affect employees’ voluntary turnover or intent to quit [67,68]. Our results also align with meta-analyses showing that many individual and organizational-level variables influence voluntary turnover [15,59,69]. More precisely, this two-lag study is among the first to analyze, through the lens of COR theory, the impact of some perceptual variables—perceived procedural justice, job insecurity, and job embeddedness—at Time 1 on voluntary turnover decisions at Time 2. So far, researchers have almost exclusively focused on the study of intent to leave through cross-sectional research design, and have not addressed the need to focus on actual turnover [15]. Our analysis also contributes to improving knowledge about preventing voluntary turnover in a particular organizational context (Bolt et al., 2022), during furlough periods, often adopted by employers to save costs and reduce layoffs.

Concerning the existing knowledge on furloughs, our results confirm COR as a relevant theoretical lens to explain the potential decoupling effects of furloughs on voluntary turnover decisions. As expected, procedural justice perceptions concerning furlough management act as a resource, reducing employees’ likelihood of quitting their jobs. In addition, as expected, job insecurity threatens furloughed employees’ resources, leading them to leave their jobs. The intrinsic or instrumental value of furloughed employees’ job embeddedness as a resource tends to act as a buffer and reduce the negative impact of perceived injustice in furlough management, reducing their motivation to leave their employer to gain resources elsewhere. Or, again, job embeddedness, a resource, tends to increase the positive impact of fair perceptions in furlough management, motivating them to stay with their employers to avoid resource losses. However, furloughed employees’ job embeddedness does not significantly buffer or reduce the positive impact of their perceived job insecurity on their decision to quit to gain resources elsewhere. This latter result might be understood in light of previous research. Bellairs et al.’s qualitative study [2] with furloughed employees described how the pandemic altered the structural and relational situation in organizations. Their respondents expressed deep uncertainties that instilled potential dignity threats through fears and anxieties, namely, concerns about returning to work, how social distancing would impact working practices, their future professional role, as well as public health and personal safety. Job embeddedness, and related inherent personal perseverance, persistence, and propensity to muster a positive response to furlough, appear insufficient in reducing the tendency of insecure furloughed employees to leave their jobs. Unsurprisingly, a focus on fear, anxiety, and health and safety was the main shaping mechanism for employee perceptions during the COVID-19 pandemic [9,70].

### 4.2. Practical Implications

Our results have implications for many workforce reduction initiatives (other than furloughs) during times of crisis. Employers and policymakers must know the potential impacts of workforce reduction decisions on turnover [71]. Furloughs feed injustice perceptions and job insecurity that increase the chance of losing employees and, most likely, the most competent ones. Our results confirm that proper procedures and good implementation allow furloughed employees to have better procedural justice perceptions, preventing them from quitting. Since management can control how furlough policies or programs are decided and implemented, they should ensure that justice or fairness (equity- or equality-based models) is prioritized and effectively communicated before, during, and after implementing a furlough policy. Management has the discretion to manage furloughs and adopt strategic human resource management practices to moderate employees’ affective responses to furloughs, which might reduce their chances of quitting [2]. A wrong cutback strategy or a poor implementation could also have negative consequences beyond employees and their turnover, including alienated clients, reputational damages, and an inability to return to pre-crisis performance [9]. The effective execution of a furlough within an organization relies on immediate supervisors. Employers should train supervisors to distribute goals and tasks appropriately and communicate effectively to enhance perceptions of fairness.

Similarly, supervisors play a role in providing personal encouragement and treating employees with dignity and respect. Moreover, optimizing the involvement of employees in decision-making in furlough processes can increase their sense of control and keep them informed about what is happening. Our results confirm that furloughed employees’ job insecurity is a source of stress [16,41]. Employers should do their best to minimize it by adopting, for example, an open and trust-based communication strategy [43] to reduce stress. Clear and transparent communication about achievements, possible organizational changes, or goals can reduce job insecurity perception during furloughs.

Finally, the study innovates and contributes to knowledge by highlighting the ubiquitous influence of furloughed employees’ embeddedness on the relationship between their procedural justice perception and their turnover decisions. Job embeddedness enables managers to retain furloughed employees and helps them understand why they stay voluntarily [72]. Employers should manage employees by building their organization and community embeddedness through links, fit, and increased sacrifices [48] because it buffers a potential lack of procedural justice perception in managing furloughs. Furloughed employees are more likely to stay if they perceive they have good relationships at work, that they fit in at work, and that leaving will come with high costs and lead to a loss of resources.

### 4.3. Limitations and Future Studies

This study has some limitations that allow us to identify future research avenues. One limit is related to the self-reported and perceptual nature of the data. Although we found that common method bias was low and we collected data at two-time points, future research could further reduce errors in parameter estimation by combining data with multiple sources (such as a manager or coworker) and at different points in time. It is impossible to conclude how the attitudes of furloughed employees changed before and after the pandemic, since data collection in T1 began during the lockdown period while many participants were already experiencing the impacts of the pandemic. Future research can focus on the temporal evolution of job insecurity and justice, as they may vary over time [52,73,74].

Additionally, the validity of our findings may be limited by the fact that an external survey company collected all data. However, some researchers agree that survey firms produce data that displays good test–retest reliability, internal consistency, factorial stability, and relationships that are very similar to those found among data directly collected by researchers [53,75,76]. Further, our results are not necessarily generalizable, given that we collected data in Canada, where a federal program allowed many organizations to engage in procedures for furloughing workers. Future research should be conducted in other contexts and consider some organizational and sector characteristics that affected furloughed employees’ attitudes and behaviours [1]. It could be interesting to investigate the potential turnover contagion process during a furlough period [77]. Finally, if one contribution of this study is to have analyzed the effect of feelings of insecurity and inequity felt by furloughed employees on their decisions to voluntary turnover, researchers could rather investigate its impacts on the involuntary turnover of volunteers. Indeed, according to COR, previous resource losses can make individuals more susceptible to job loss. After experiencing injustice and job insecurity, furloughed employees’ performance when they come back to work can be impaired and lead to losing their job involuntarily. Similarly, and as recommended by Bolt et al. [15], the impact of furloughs on those who leave or stay should be meaningfully assessed using a differentiated approach that identifies functional and dysfunctional, avoidable and unavoidable, turnover and retention.

## 5. Conclusions

Based upon COR theory, we conducted a two-time point design among furloughed employees during the 2020 COVID-19 lockdown in Canada. Our results confirm that their perceived procedural justice of furlough management and job insecurity (measured in Time 1) are related to their subsequent voluntary turnover decisions (Time 2). In addition, furloughed employees’ job embeddedness (measured in Time 1) contributes to motivating them to stay with their employer, whatever their procedural justice perceptions of furlough management. Many employer-led practices can prevent and reduce the negative impacts of furloughs on voluntary turnover decisions, which have financial and human costs at the individual, organizational, social, and societal levels.

## Figures and Tables

**Figure 1 ijerph-20-05664-f001:**
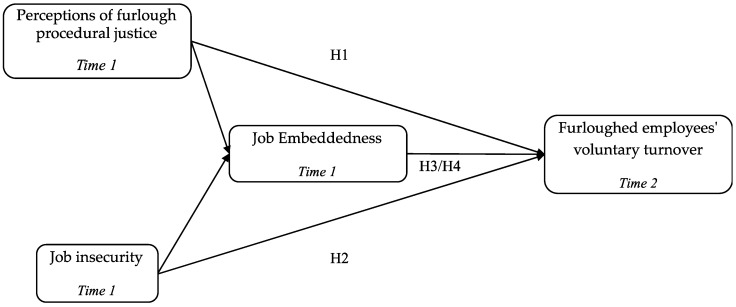
Research model.

**Table 1 ijerph-20-05664-t001:** Correlations among research variables.

	1	2	3	4	5	6	7	8	9
1. Voluntary turnover decision *T2*	1								
2. Perceived job embeddedness *T1*	−0.091	1							
3. Perceived job insecurity *T1*	0.149 *	−0.057	1						
4. Perceived justice in furlough management *T1*	−0.181 **	0.253 **	0.001	1					
5. Perceived employability *T1*	−0.245 **	0.135 **	−0.134 **	0.153 **	1				
6. Age	0.046	−0.017	0.150 **	−0.069	−0.063	1			
7. Gender	−0.018	0.037	−0.019	0.043	0.002	−0.237 **	1		
8. Education	0.044	0.021	0.058	0.038	−0.043	0.006	0.018	1	
9. Heures semaine	0.055	0.040	0.088 *	−0.010	−0.051	−0.019	0.001	−0.098 *	1

Note. * *p* < 0.05, ** *p* < 0.01.

**Table 2 ijerph-20-05664-t002:** Results of logistic regression analyses.

Voluntary Turnover Decision	Estimate	OR	t	*p*
Perceived justice in furloughs management at Time 1	−0.40	0.668	−3.0	0.000
Perceived job insecurity at Time 1	0.23	1.26	2.5	0.014
Control variables:				
Perceived employability	−0.45	0.636	−3.2	0.001
Age	0.01	1.1	0.63	0.529
Gender	0.04	1.0	0.15	0.884
Education	0.04	1.4	0.31	0.754

**Table 3 ijerph-20-05664-t003:** Results of job embeddedness indirect effects.

	Effect on Voluntary Turnover	Odds Ratio for Voluntary Turnover
	Estimate (95% CI)	aOR (95% CI)
Job insecurity (x) → Job embeddedness (M) → Voluntary turnover (Y)	0.002 (−0.013, 0.017)	1.002 (0.987, 1.017)
Procedural justice(x) → Job embeddedness (M) → Voluntary turnover (Y)	−0.048 (−0.083, −0.021)	0.953 (0.921, 0.979)

Note. CI = 95% confidence interval.

## Data Availability

The data presented in this study are available on request from the corresponding author.

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
