# Peer review of "Furloughed Employees’ Voluntary Turnover: The Role of Procedural Justice, Job Insecurity, and Job Embeddedness"

_ijerph, 2023, doi:10.3390/ijerph20095664_

Round 1

Reviewer 1 Report

The article deals with the furloughed employees' voluntary turnover decision. Although this issue may be interesting to follow, I have a lot of doubts about the quality of this paper.

Abstract should be more informative, with the clearly stated motivation of the paper and clearly presented findings.

Introduction suffers from lack of transparent and comprehensible context for the discussion in the body of the paper. The motivation of the paper should be strengthen as well as research gap and contributions.

Theoretical part requires improvement. There is a noticeable lack of in-depth critical analysis of literature, which indicates insufficient development of the main body of the paper. The hypotheses development is insufficient, with lack of proper adjustment on the base of literature. I do recommend rewriting this part of the paper in a comprehensive, complex and logic way. There are also a lot of statements not related to the paper.

The methodology part is very unclear and should be rewritten to be more related to the main body of the paper. At this moment, the presentation of the research method is difficult to follow, confusing, with a lot of statements not related directly to the paper. The description of the variables used in the empirical research is weak, presented in a very incoherent way. Moreover, the Authors do not sufficiently develop issues related to the applied methods. The authors put in this part of the paper the whole paragraph related to suggestion how to write methodology part: ‘The Materials and Methods should be described with sufficient details to allow others to replicate and build on the published results. Please note that the publication of your manuscript implicates that you must make all materials, data, computer code, and protocols associated with the publication available to readers. Please disclose at the submission  stage any restrictions on the availability of materials or information. New methods and  protocols should be described in detail while well-established methods can be briefly described and appropriately cited’. This is unacceptable.

I am also not convinced about the Authors’ presentation of the results. This part of the paper needs substantial changes. I do believe that the Authors did a lot of work to carry out the research, but the presentation of the results is presented in a very unclear way and confusing. The discussion should be more related to the aim of the paper. There are also a lot of statements not directly related to the aim of the paper.

Conclusions need substantial changes as well. At this moment, this part of article does not provide a neat summary in relation to the aim of the paper.

Author Response

REVIEW 1

We hope to respond with the greatest clarity to the questions and recommendations raised. We followed those closely and made all the corrections and additions requested. 

In yellow: all text sections that have been considerably revised in terms of form or content.

In blue: all the references added to the list of references.

The article deals with the furloughed employees' voluntary turnover decision. Although this issue may be interesting to follow, I have a lot of doubts about the quality of this paper. Abstract should be more informative, with the clearly stated motivation of the paper and clearly presented findings.

Answer:

Thank you for recognizing that our manuscript is interesting to follow. We rewrote many parts of our paper to improve its quality. Among others, we have completely revised the abstract, and now it is clearer and better illustrates the objectives and results of our article.

Introduction suffers from lack of transparent and comprehensible context for the discussion in the body of the paper. The motivation of the paper should be strengthened as well as research gap and contributions.

Answer:

We agree with your comment. We have completely rewritten our introduction to add more contextual information and definitions, highlighting the research gap and contributions. The common thread is also clearer. Thank you for getting us to make this improvement.

Theoretical part requires improvement. There is a noticeable lack of in-depth critical analysis of literature, which indicates insufficient development of the main body of the paper. The hypotheses development is insufficient, with lack of proper adjustment on the base of literature. I do recommend rewriting this part of the paper in a comprehensive, complex and logic way. There are also a lot of statements not related to the paper.

Answer:

Thanks for this comment. We rephrased this section to better describe the links between our theoretical framework and all our hypotheses. We also present the figure of our model right before our hypotheses. We extract the points that are not relevant or related to our hypotheses.

The methodology part is very unclear and should be rewritten to be more related to the main body of the paper. At this moment, the presentation of the research method is difficult to follow, confusing, with a lot of statements not related directly to the paper. The description of the variables used in the empirical research is weak, presented in a very incoherent way. Moreover, the Authors do not sufficiently develop issues related to the applied methods. The authors put in this part of the paper the whole paragraph related to suggestion how to write methodology part: 'The Materials and Methods should be described with sufficient details to allow others to replicate and build on the published results. Please note that the publication of your manuscript implicates that you must make all materials, data, computer code, and protocols associated with the publication available to readers. Please disclose at the submission stage any restrictions on the availability of materials or information. New methods and protocols should be described in detail while well-established methods can be briefly described and appropriately cited'. This is unacceptable.

Answer:

As recommended, we rewrote this section almost entirely to improve its clarity and common thread. We added much more information on participants, measures, and how we conducted data analyses. Many thanks for our advice that help demonstrate our study's rigor. Regarding the availability of information or materials, we agree to make these types of information available.

I am also not convinced about the Authors' presentation of the results. This part of the paper needs substantial changes. I do believe that the Authors did a lot of work to carry out the research, but the presentation of the results is presented in a very unclear way and confusing.

Answer:

You are right. We reformulated our results in light of our model, which now associates each arrow with one hypothesis. In giving more details concerning results tables, we now have an answer to your legitime advice. Thanks.

The discussion should be more related to the aim of the paper. There are also a lot of statements not directly related to the aim of the paper.

Answer:

We have further developed this section by discussing the implications of our study for theory and practice. We put out all ideas not directly related to its aim. We have greatly enriched the content by deepening the implications of our findings and making more connections with other theoretical or empirical writings.

Conclusions need substantial changes as well. At this moment, this part of article does not provide a neat summary in relation to the aim of the paper.

Answer:

Thanks for this comment. We rewrote our conclusion to summarize our study's goal, results, and main contributions.

In closing, we went beyond the demands. A review of our article also allowed us to improve the common thread, to remove repetitions or less relevant elements. We have greatly improved the sections of our manuscript (introduction, literature review, method and discussion). We have also removed references to add more in some places. We hope you will appreciate our corrections and additions.

Reviewer 2 Report

Review report--ijerph-2225970-peer-review-v1

l   As most countries and regions lift quarantines or restrictions in response to COVID-19, the adverse impact of the pandemic on people is likely to continue for some time. In the post-pandemic era, these issues need to be properly addressed. In this way, mankind can get out of the predicament caused by the epidemic as soon as possible. During the COVID-19 pandemic, many businesses or companies had to lay off employees in various ways, such as working shifts, which forced some employees to face unpaid leave. The research of this manuscript has certain significance and value. The ideas and methods of research are suitable, and the structure of the manuscript is more reasonable.

But before it could be published, The following comments and suggestions are need to be clarified, explain and revised :

1.     The abstract needs to be rewritten. It requires a brief description of the background, motivation, and purpose of the research. Conclusions and recommendations should not be omitted either.

2.     Line 219~225: This text is a description of a section in the template, please delete it.

3.     What industries did the respondents come from? Have the authors investigated this? For example, traditional industries may have to be closed due to the epidemic, while Internet companies have also been affected by the epidemic, but their work content can be carried out online.

4.     With 70 percent of respondents between the ages of 25 and 44, this group is the backbone of the business. As the pandemic gradually ends, do the authors continue to track the recent status of these respondents? In other words, as life gradually returns to the state it was in 2019, will unpaid leave or so-called short-term leave still be common for employees? And will they still have an impact?

5.     Chapter 4: Authors need to moderately reduce the frequency of re-citations. We look forward to seeing the authors' thoughts and perspectives.

6.     As just mentioned, the conclusions may need to be further adjusted. It reflects the findings of this study. So how can these findings be better used when the pandemic is over? During the economic recovery phase, employees may value job stability more, and these people are also very important to the business.

Author Response

REVIEW 2

We hope to respond with the greatest clarity to the questions and recommendations raised. We followed those closely and made all the corrections and additions requested. 

In yellow: all sections of the text that have been considerably revised in terms of form or content.

In blue: all the references added to the list of references.

As most countries and regions lift quarantines or restrictions in response to COVID-19, the adverse impact of the pandemic on people is likely to continue for some time. In the post-pandemic era, these issues need to be properly addressed. In this way, mankind can get out of the predicament caused by the epidemic as soon as possible. During the COVID-19 pandemic, many businesses or companies had to lay off employees in various ways, such as working shifts, which forced some employees to face unpaid leave. The research of this manuscript has certain significance and value. The ideas and methods of research are suitable, and the structure of the manuscript is more reasonable.

But before it could be published, The following comments and suggestions are need to be clarified, explain and revised:

Answer:

Thank you for recognizing the relevance and quality of our study, the research method, and manuscript structure. We have improved it a lot based on all reviewers' comments. We are grateful for your advice.

The abstract needs to be rewritten. It requires a brief description of the background, motivation, and purpose of the research. Conclusions and recommendations should not be omitted either.

Answer:

You are right. We have rewritten the abstract to make it more exhaustive, as requested.

Line 219~225: This text is a description of a section in the template, please delete it.

Answer:

Thank you for pointing us toward this error. We have deleted it.

What industries did the respondents come from? Have the authors investigated this? For example, traditional industries may have to be closed due to the epidemic, while Internet companies have also been affected by the epidemic, but their work content can be carried out online.

Answer:

Participants from several business sectors represent the sample. We have added more information regarding this variable in the sample description.

As part of our approach, we solicited individuals who were furloughed. In this context, these people could not continue working because of the pandemic despite the possibility of working remotely.

With 70 percent of respondents between the ages of 25 and 44, this group is the backbone of the business. As the pandemic gradually ends, do the authors continue to track the recent status of these respondents? In other words, as life gradually returns to the state it was in 2019, will unpaid leave or so-called short-term leave still be common for employees? And will they still have an impact?

Answer:

Our study targeted furloughed employees during the period of confinement during the pandemic. However, unpaid leave has historically been frequently used in the event of financial difficulties, and everything indicates that it will still be used in the future (see our references list). Therefore, our results may be relevant for future implemented furloughs and other contexts.

Chapter 4: Authors need to moderately reduce the frequency of re-citations. We look forward to seeing the authors' thoughts and perspectives.

Answer:

We have reread our text with this advice in text. We have attempted to retain the citations for definitions only and the description of the theoretical perspective. The text of the discussion also responds to this comment.

As just mentioned, the conclusions may need to be further adjusted. It reflects the findings of this study. So how can these findings be better used when the pandemic is over? During the economic recovery phase, employees may value job stability more, and these people are also very important to the business.

Answer:

Thanks for this comment. We have further developed our study's theoretical and practical contributions for contexts beyond the pandemy.

In closing, we went beyond the demands. A review of our article also allowed us to improve the common thread, to remove repetitions or less relevant elements. We have greatly improved the sections of our manuscript (introduction, literature review, method and discussion). We have also removed references to add more in some places. We hope you will appreciate our corrections and additions.

Reviewer 3 Report

dear authors, nice presentation of the article. there are some points I would like to mention.

1. i believe you should use the term "employee or staff turnover".

2. the term "furloughs" should be clearly explained by giving examples

3. the text at line 219 to 224 should be deleted.  

Author Response

REVIEW 3

We hope to respond with the greatest clarity to the questions and recommendations raised. We followed those closely and made all the corrections and additions requested. 

In yellow: all sections of the text that have been considerably revised in terms of form or content.

In blue: all the references added to the list of references.

Dear authors, nice presentation of the article. there are some points I would like to mention.

Thank you for recognizing the quality of our manuscript. We have improved it a lot based on all reviewers' comments. We are grateful for your advice.

I believe you should use the term "employee or staff turnover".

Answer:

We revised our paper to mostly use "employee or staff turnover", a term more used in the literature. We have used other terms in places (leave) for form reasons only.

The term "furloughs" should be clearly explained by giving examples

Answer:

You are right. Our text now clearly defines this term and gives examples at the beginning of the introduction. Thanks for this comment.

The text at line 219 to 224 should be deleted.

Answer:

Thank you for pointing us toward this error. We have deleted this part of the text.

In closing, we went beyond the demands. Reviewing our article also allowed us to improve the common thread, removing repetitions or less relevant elements. We have greatly enhanced the sections of our manuscript (introduction, literature review, method, and discussion). We have also removed references to add more in some places.

Reviewer 4 Report

First of all, I think the study is of interest within the scope of understanding how work environment such as injustice procedures and job insecurity, is related to turnover. The empirical findings are solid. I think you may elaborate some more on this in your conclusion. It´s important. But to frame your results you need to read up some more on the knowledge field. some of the references to the knowledge field is quite old. There has been much research on turnover the last decades. For example, it should be important to read up on this field. Maybe by starting with a review, for example Trees Bolt et al., (2022) A century of labour turnover research: A systematic literature review. You need to also discuss your findings in reference to earlier research. What is new? What is in line? 

My advice is for you to change the theoretical lens. It is not clearly showed in the text how the theory you say you use is linked to the findings. Your findings show that furloughed employees’ job insecurity and perceived furlough management justice are related to their turnover decision. Also furloughed employees' job embeddedness has an indirect mediator effect on the relationship between their perceived justice in furlough management and their turnover decision. I really don´t se how COR may be of use analyzing this result.

Furloughs management is often analyzed with different theory, such as equity theory or social exchange theory. Why choose resources theory (COR) that only helps you explain individual level when the results show strong links to injustice and job insecurity? Reading your findings that stress the importance of correlations to justice procedure, is hard to see how COR theory is the best match for understanding the results. Maybe a theory on organizational justice or social exchange theory would be more accurate? 

If you work with improvements theoretically, the results will be more lucid and stringent.

Author Response

REVIEW 4

First of all, I think the study is of interest within the scope of understanding how work environment such as injustice procedures and job insecurity, is related to turnover. The empirical findings are solid. I think you may elaborate some more on this in your conclusion. It's important. But to frame your results you need to read up some more on the knowledge field. Some of the references to the knowledge field is quite old.

Answer:

Thank you for recognizing the relevance of our study and its findings in the field of knowledge on turnover. We have improved it a lot based on all reviewers' comments. We are grateful for your advice and we have updated our references. Among other things, we cited Trees Bolt et al., (2022) , A century of labour turnover research: A systematic literature review, which recommended studying turnover in specific contexts such as crises, difficult situations, etc. We also enhanced our discussion to better demonstrate our contributions to knowledge on furloughs and turnover as well as to practitioners.

There has been much research on turnover the last decades. For example, it should be important to read up on this field. Maybe by starting with a review, for example Trees Bolt et al., (2022) A century of labour turnover research: A systematic literature review. You need to also discuss your findings in reference to earlier research. What is new? What is in line?

Answer:

Thank you very much for this reference. We refer to it in many places in our paper. This recent literature review confirms the relevance of our study. Our discussion points out what is new and what is in line with prior works.

My advice is for you to change the theoretical lens. It is not clearly showed in the text how the theory you say you use is linked to the findings. Your findings show that furloughed employees' job insecurity and perceived furlough management justice are related to their turnover decision. Also furloughed employees' job embeddedness has an indirect mediator effect on the relationship between their perceived justice in furlough management and their turnover decision. I really don't se how COR may be of use analyzing this result. Furloughs management is often analyzed with different theory, such as equity theory or social exchange theory. Why choose resources theory (COR) that only helps you explain individual level when the results show strong links to injustice and job insecurity? Reading your findings that stress the importance of correlations to justice procedure, is hard to see how COR theory is the best match for understanding the results. Maybe a theory on organizational justice or social exchange theory would be more accurate? If you work with improvements theoretically, the results will be more lucid and stringent

Answer:

We analyse this advice that was not shared by other reviewers. We have decidesd to rewrite the literature review section and better articulated our research hypotheses with the assets of the COR theory. We hope that presenting the hypotheses to the light of this theory seems more convincing to you. Obviously, in the discussion, we also put forward thatour results are consistent with other theoretical perspectives such as organizational justice and the theory of social exchange. Our conclusion also echoes more past work on turnover and our contribution to this field of knowledge. Thank you very much for your opinions.

In closing, we went beyond the demands. Reviewing our article also allowed us to improve the common thread, removing repetitions or less relevant elements. We have greatly improved the sections of our manuscript (introduction, literature review, method, and discussion). We have also removed references to add more in some places.

Round 2

Reviewer 1 Report

The quality of the article was significantly improved by the Authors' changes to the text.

Author Response

We are grateful for your comments and advice.

Regards,

The authors

Reviewer 4 Report

The text is now much more lucid and relevant to the field 

Author Response

(The authors gave the same response as above.)
